# Femoral Component Rotation in Total Knee Arthroplasty Using a Tibia-First, Gap-Balancing, “Functional Alignment” Technique

**DOI:** 10.3390/jcm11226680

**Published:** 2022-11-10

**Authors:** Constantin Dlaska, Petros Ismailidis, Kenji Doma, Benjamin Brandon, Matthew Wilkinson, Kaushik Hazratwala

**Affiliations:** 1The Orthopaedic Research Institute of Queensland (ORIQL), Pimlico, Townsville, QLD 4812, Australia; 2Division of Tropical Health and Medicine, College of Health Care Sciences, James Cook University, Douglas, QLD 4811, Australia; 3Department of Orthopaedics and Traumatology, University Hospital of Basel, Spitalstrasse 21, 4031 Basel, Switzerland; 4School of Medicine, University of Tasmania, Medical Science Precinct, 17 Liverpool St., Hobart, TAS 7000, Australia; 5Division of Medicine and Dentistry, James Cook University, Douglas, QLD 4811, Australia

**Keywords:** TKA, total knee arthroplasty, rotational alignment, kinematic alignment, functional alignment, gap balancing

## Abstract

Background: The purpose of this study was to describe the femoral component rotation in total knee arthroplasty (TKA) using a tibia-first, gap-balancing, “functional alignment” technique. Methods: Ninety-seven patients with osteoarthritis received a TKA using computer navigation. The tibial resection was performed according to the kinematic alignment (KA) principles, while the femoral rotation was set according to the gap-balancing technique. Preoperative MRIs and intraoperative resection depth data were used to calculate the following rotational axes: the transepicondylar axis (TEA), the posterior condylar axis (PCA) and the prosthetic posterior condylar axis (rPCA). The angles between the PCA and the TEA (PCA/TEA), between the rPCA and the PCA (rPCA/PCA) and between the rPCA and the TEA (rPCA/TEA) were measured. Data regarding patellar maltracking and PROMs were collected for 24 months postoperatively. Results: The mean PCA/TEA, rPCA/TEA and rPCA/PCA angles were −5.1° ± 2.1°, −4.8° ± 2.6° and −0.4° ± 1.7°, respectively (the negative values denote the internal rotation of the PCA to the TEA, rPCA to TEA and rPCA to PCA, respectively). There was no need for lateral release and no cases of patellar maltracking. Conclusions: A tibia-first, gap-balancing, “functional alignment” approach allows incorporating a gap-balancing technique with kinematic principles. Sagittal complexities in the proximal tibia (variable medial and lateral slopes) can be accounted for, as the tibial resection is completed prior to setting the femoral rotation. The prosthetic femoral rotation is internally rotated relative to the TEA, almost parallel to the PCA, similar to the femoral rotation of the KA-TKA technique. This technique did not result in patellar maltracking.

## 1. Introduction

The axial alignment (rotation) of the femoral component in mechanically aligned total knee arthroplasty (MA-TKA) follows the two main surgical techniques, measured resection and gap balancing [1]. Using the measured-resection technique, the rotation of the femoral component is decided based on predefined anatomical parameters, namely, the transepicondylar axis (TEA), the antero-posterior axis (APA) or a line three degrees externally rotated relative to the posterior condylar axis (PCA) [2]. The flexion gap can be rectangular or trapezoidal. Studies have shown a much higher incidence of femoral condylar lift-off with this technique [3] and potentially a wide disparity between a balanced flexion gap and any of these axes [4,5]. Using the gap-balancing technique, the knee is first balanced in extension. The rotation of the femoral component is set according to the soft tissue tension in 90° of flexion. This achieves equal balance in flexion and extension, but the rotation of the femoral component varies freely within the restrictions of the soft tissue structures. Several studies have reported the femoral rotation with gap-balancing techniques in MA-TKAs and found it to have a wide variation with an average similar to that of the measured resection MA-TKA, slightly externally rotated in relation to the PCA [6,7,8,9,10]. Excessive medial release and varus proximal tibial resection, however, can result in a significant internal rotation of the femoral component [6].

Kinematic alignment TKA(KA-TKA) follows a measured resection philosophy in order to reconstruct the native tibiofemoral articular surfaces and therefore restore the native laxity of the knee [11]. Femoral rotation is decided according to the chondral wear pattern of the posterior condyles, and thus the rotational alignment of the femoral component is essentially parallel to the PCA. Studies assessing the rotation of the femoral component in KA-TKA show that the femoral component is internally rotated in relation to the APA and the TEA [2,12]. Recent studies have shown that this results in a more anatomic restoration of the trochlear sulcus location and sulcus angle [13,14]. 

Flexion balance in TKA is influenced by the rotational alignment of the femoral component, as well as by the coronal and sagittal alignment of the tibial component. Regarding the sagittal tibial alignment, the native knee has a wide variation in constitutional posterior tibial slope (PTS) as well as, often, a differential PTS between the medial and lateral sides [15,16,17]. Prosthetic design does not allow for differential PTS between the medial and lateral sides, and surgical techniques tend to plan for a fixed target PTS. These factors can complicate the flexion balance with a femur first surgical technique.

In order to address this problem, a tibia-first, gap-balancing, “functional alignment” technique was developed based on both kinematic and gap-balancing philosophies [18]. Tibial and distal femoral resection is performed according to the KA-TKA principles. Contrary to KA-TKA, the femoral rotation is decided according to the ligament tension in 90° of flexion in order to achieve a balanced knee joint. Improved patient outcomes with a similar technique when compared with mechanical alignment has been reported [16].

The aim of this study was to examine the rotational alignment as well as the possible patellofemoral complications resulting from a tibia-first, gap-balancing, “functional alignment” technique. The primary hypothesis was that the axial alignment of the femoral component would be parallel to the PCA. The secondary hypothesis was that the variation from the PCA would be explained by the medial-to-lateral PTS differential.

## 2. Materials and Methods

### 2.1. Ethics

This study was performed in line with the principles of the Declaration of Helsinki. Approval was granted by the local Ethics Committee (Mater Health Services MHS20180821-03, Human Research Ethics Committee James Cook University, Townsville, QLD, Australia H8762).

### 2.2. Participants

From April 2016 to July 2018, 156 consecutive patients with knee OA planned for a TKA by one senior surgeon with extensive experience in MA and KA-TKA at a single institution were screened for possible inclusion. The inclusion criterion was patients undergoing primary TKA for the treatment of knee osteoarthritis with an available preoperative MRI. The exclusion criteria were ligamentous instability requiring a constrained condylar knee TKA design (presence of tibial intramedullary stem), patient refusing to give consent to the study and MRI quality not allowing to calculate the anatomical axes used as outcome measures.

### 2.3. Surgical Technique

All participants received a cemented TKA with a rotating bearing tibial insert and patella resurfacing (Attune TKA System, DePuy Synthes, Warsaw, IN, USA). A tibia-first, gap-balancing, “functional alignment” technique was applied in all cases using an imageless computer navigation system (BrainLAB^®^ VectorVision^®^ Knee 3 Navigation System, Brainlab, Munich, Germany).

This surgical technique [18] combines the kinematic and gap-balancing approaches using computer navigation. Radiographic pre-operative assessment is utilized from long-leg X-Rays, low-dose CT computer program analysis, EOS whole-body scans or MRIs (for the present study, all patients received a preoperative MRI). Hip–Knee–Ankle angle (HKAA), mechanical Lateral–Distal–Femoral angle (mLDFA), mechanical Medial–Proximal–Tibial angle (MTPA) as well as PTS for medial and lateral compartments are recorded. Intraoperatively, the prearthritic constitutional HKAA in extension and 90° flexion is defined by stressing the collateral ligament of the affected arthritic side until tensioned. The tibial resection is performed first, to match the prearthritic MPTA and PTS. In cases where the medial and lateral PTS are different, whichever PTS (medial or lateral) is largest is selected as the target PTS, with a value of 9° defined as maximum. The distal femoral resection is performed according to the prearthritic mLDFA and in order to restore the prearthritic HKAA in extension. The knee is then flexed at 90°, and the distal anterior and posterior femoral cuts are performed according to the gap-balancing principles in order to restore the prearthritic tibiofemoral alignment in 90° flexion with a balanced flexion gap. The trial prosthesis is inserted, and kinematics tested, including patellofemoral joint tracking. The definitive prosthesis is inserted, and routine closure commenced.

Posterior condyle resection data were recorded at the time of the surgery using computer navigation read-out data. The resection was measured in millimetres of posterior condyle resection. According to the KA-TKA technique, the resection depth (bone and cartilage) matches the depth of the prosthesis. The compartment of the knee joint having the higher cartilage loss normally has a smaller resection to accommodate for cartilage wear. The opposite, i.e., higher resection at the worn than at the unworn compartment, was defined as reverse resection. TKAs with reverse resections were isolated and separately analysed.

### 2.4. Clinical Outcomes

Patient-reported outcome measures (PROMs) were collected for all patients pre-operatively and post-operatively at the 6 weeks, 6 months, 1 year and 2 year marks, i.e., the Western Ontario and McMaster Universities Osteoarthritis Index (WOMAC) and the Knee Society Score (KSS).

To assess for a possible patellofemoral joint malalignment, the need of intraoperative lateral release as well as postoperative patellar instability or dislocation were recorded.

### 2.5. Radiological Outcomes

A preoperative MRI was performed on all patients. All MRIs were obtained using a 3 Tesla, high-resolution, lower extremity imaging coil, and axial, T2-weighted, fast-spin, fat saturation images with slice thickness of 6 mm were analysed (Siemens Healthcare, Erlangen, Germany).

The PTS was measured separately for each compartment according to the technique described by Hashemi et al. [15]. The technique for measuring the rotation of the femoral component based on MRI images was previously described by Park, 2014 [12]. In a split-screen mode, with the cross-reference line tool activated, sagittal and axial images were selected. The sagittal image which showed the largest diameter of the lateral femoral condyle was selected. A best-fit circle tool was used to identify the position of the epicondyle. The corresponding cross-sectional axial image was selected in the adjacent screen. All measurements were performed on this screen (Figure 1).

The pre-arthritic PCA was defined as the tangent to the posterior condyles after allowing for up to 2 mm correction in case of cartilage loss, as evident on the MRI. The TEA was defined as the line connecting the most prominent medial and lateral aspects of the respective epicondyles. Using the measured navigated resection depth data, the respective distance from the posterior condyles was marked, and a point signifying the depth of the resection was defined on each condyle (Figure 2a). The line connecting these points was the resected posterior condylar axis (rPCA, Figure 2b). The rPCA corresponds to the placement of the femoral component and is parallel to the implant’s posterior condylar axis. The angles between the PCA and the TEA (PCA/TEA angle), between the rPCA and the TEA (rPCA/TEA angle) and between the rPCA and the PCA (rPCA/PCA angle) were measured in each case (Figure 3). All assessments were performed independently by two investigators (CD, BB) using the same technique on the same axial image that was selected by the first investigator (CD).

### 2.6. Statistical Analysis

The measure of central tendency and dispersion was reported as mean ± standard deviation for all data and analysed using the Statistical Package of Social Sciences (SPSS, version 25; IBM Corp., Armonk, NY, USA) software. According to the Shapiro–Wilks test, the dependent parameters departed from the norm; thus, all data were log-transformed (log10[x]) prior to the statistical analyses. The inter-observer reliability and measurement error between two independent observers for the PCA/TEA, rPCA/TEA and rPCA/PCA ratios were determined using the intra-class correlation coefficient (ICC, 2-way mixed, 95% confidence intervals) and the coefficient of variation (CV) with associated 95% confidence intervals. A one-way repeated-measures analysis of variance (ANOVA) was used to examine the differences in the KSS and WOMAC measures between the different time points.

## 3. Results

### 3.1. Patient Demographics

Out of 156 TKAs screened, 97 TKAs in 83 patients (54% female, 46% male) met the inclusion criteria. The mean (±standard deviation SD) age was 66 ± 10 years, BMI 30 ± 6 kg/m^2^.

### 3.2. Clinical and Radiological Outcomes

Patient-reported outcome scores are reported in Table 1. There was a significant improvement in the KSS and WOMAC scores over a twelve-month post-operative period (*p* < 0.01). Over this follow-up period, no cases of patella instability or dislocation were recorded. There were no patients who required a lateral release at time of surgery.

The mean (±SD) PCA/TEA angle was −5.1 ± 2.1°, the rPCA/TEA angle was −4.8 ± 2.6°, and the rPCA/PCA angle was −0.4 ± 1.7° (the negative values denote the internal rotation of the PCA to the TEA, rPCA to TEA and rPCA to PCA, respectively). The inter-rater reliability was high for the measurements of the rPCA/PCA angle (0.81 [CI 0.70–0.87]) and low for the PCA/TEA (0.6 [CI 0.42–0.74]) and rPCA/TEA (0.63 [CI 0.44–0.75]) measurements.

Eight patients were identified as having reverse resections of the posterior condyles. Table 2 presents the amount resected from the posterior condyles as well as the PTS of the medial and lateral compartments for these patients. All patients except one had increased posterior slope on the predominantly affected side.

## 4. Discussion

The present study demonstrates that the rotational alignment of the prosthetic femoral component using a tibia-first, gap-balancing, “functional alignment” technique is almost parallel to the PCA and internally rotated relative to the TEA. Consistent with previous studies, the native PCA of our cohort was internally rotated relative to the TEA. No need for patellofemoral release or patellar instability/dislocation was recorded.

In MA-TKA, the TEA has been used as a surrogate for the centre-of-knee flexion and extension, and this has been incorporated into surgical techniques and modern implant designs. The internal rotation of the femoral component is avoided to prevent patellofemoral complications. According to the MA-TKA principles, the prosthetic femoral rotation of our cohort would be interpreted as an internal malrotation. A recent meta-analysis by Corona et al. [19] showed that the so-called “malrotation” of the femoral component did not correlate with a poor clinical or functional outcome. The authors concluded that careful attention to the variability of the native distal femur may contribute to the understanding of the unhappy prosthetic knee [19]. Further studies have also shown that internally rotating the femoral component does not necessarily adversely affect the patellofemoral kinematics [2] and provided evidence for the contrary; the recreation of constitutional trochlear anatomy delivers improved clinical outcomes [20,21].

KA-TKA achieves a more anatomic restoration of the trochlear anatomy [13,14]. Park et al. [12] reported that the TKAs implanted with the KA technique were on average 4° internally rotated relative to the TEA and 0.5° externally rotated relative to the PCA. This rotational alignment is very similar to the one reported in our study. The slight discrepancy may be the result of the KA technique being a purely measured resection technique and not taking into consideration the balance in flexion and the differential PTS between the medial and lateral tibial plateaus. Park et al. [12] assumed the cartilage loss to be 2 mm on the affected side and rotated the component accordingly. Furthermore, the original KA technique is a femur-first technique and therefore does not allow incorporating the advantages of a gap-balanced approach to ensure flexion balance. Contrary to the KA technique, the technique discussed in the present study orientates the rotation of the femoral component according to the gap-balancing principles after the tibial cut is performed to match the constitutional anatomy. At the same time, it adjusts the femoral rotation to correct the possible flexion gap imbalance resulting from the single PTS of the prosthetic TKA design.

The medial and lateral tibia plateaus have different PTS, the medial being more frequently larger [17]. However, the prosthetic TKA designs have a single PTS for both medial and lateral plateaus. This has an effect on the TKA balance in flexion. Using the technique described above, balance in flexion is obtained by adjusting the rotation of the femoral component. This could result in a reverse resection of the posterior femoral condyles (higher resection at the affected compartment) and therefore in a femoral component rotation deviating from the PCA. This can be explained by the differential in the PTS affecting the balance. For instance, three patients with lateral arthritis had a lateral PTS larger than the medial (Table 2). The slope resection was in all cases matched to the lateral side with a maximum of 9°. Consequently, in order to balance the knee in flexion, more bone had to be taken from the worn, affected lateral posterior femoral condyle; thus, a reverse resection and an internal rotation of the femoral component occurred. The opposite occurred in knees with medial arthritis having a medial slope larger than the lateral one. Reverse resection does not occur in the KA technique, and therefore rotational alignment is expected to be more consistently aligned to the PCA, which however, could result in a flexion imbalance.

The rotation of the femoral component is known to be significant for its implications in patellofemoral tracking. It is also a critical step in balancing the total knee replacement in flexion. Studies have shown advantages in the gap-balancing surgical philosophy to achieve this over a measured-resection approach [3,22]. Clinically, we did not record adverse patellofemoral joint events. There were no cases of patellofemoral instability or lateral release in our series. This is in agreement with recent studies looking at the patellofemoral joint in kinematically aligned knees [13,14]. Our overall PROM results were similar to those reported in the literature for the MA and KA-TKA techniques (Table 1) [23,24].

### Strengths and Limitations

The lack of a control group is an obvious limitation of this study. However, the purpose of this study was to describe the femoral rotation with this tibia-first, gap-balancing, “KA technique; therefore, a control group was not necessary. The results regarding the femoral rotation and PROMs were placed in the context of the current literature and compared with data published in different studies. The moderate size of the patient group could be a limitation. The facts that all the operations were performed by a single surgeon as well as the standardised radiological measurements were performed by two investigators are a strength of this study.

## 5. Conclusions

A tibia-first, gap-balancing, “functional alignment” technique allows the incorporation of a gap-balancing technique with kinematic principles. Sagittal complexities in the proximal tibia (variable medial and lateral slopes) can be accounted for, as the tibial resection is completed prior to setting the femoral rotation. The resulting prosthetic femoral component is internally rotated relative to the TEA and almost parallel to the PCA, similar to the femoral rotation of the KA-TKA technique. According to our experience, this is a safe, reproducible technique. This technique did not result in patellar maltracking.

## Figures and Tables

**Figure 1 jcm-11-06680-f001:**
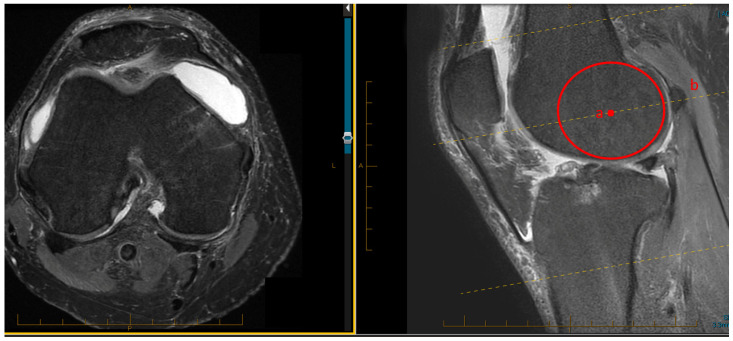
Technique for measuring the rotation of the femoral component based on MRI images: in a split-screen mode, with the cross-reference line tool (dotted yellow lines) activated, sagittal (**right**) and axial (**left**) images were selected. The sagittal image which showed the largest diameter of the lateral femoral condyle was selected. A best-fit circle tool was used to identify the position of the epicondyle (point a, red circle). The corresponding cross-sectional axial image (dotted yellow line, b) was selected in the adjacent screen. All measurements were performed on this screen.

**Figure 2 jcm-11-06680-f002:**
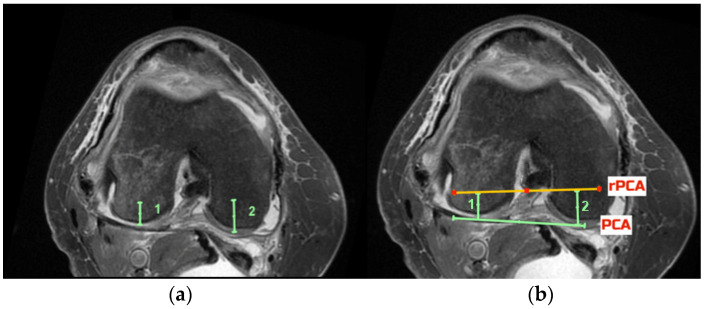
Defining the preartrhitic posterior condylar axis (PCA) and the resected posterior condylar axis (rPCA). (**a**): Using the measured navigated resection depth data, the respective distance from the posterior condyles was marked, and a point signifying the depth of the resection was defined on each condyle (in this case, distance 1: 4 mm, and distance 2: 6 mm). (**b**) The line connecting these two points was the prosthetic posterior condylar axis (rPCA, yellow). The rPCA corresponds to the placement of the femoral component and is parallel to the implant’s posterior condylar axis. The posterior condylar axis (PCA) was defined as the tangent to the posterior condyles after allowing for up to 2 mm correction in case of cartilage loss, as evident on the MRI (green line).

**Figure 3 jcm-11-06680-f003:**
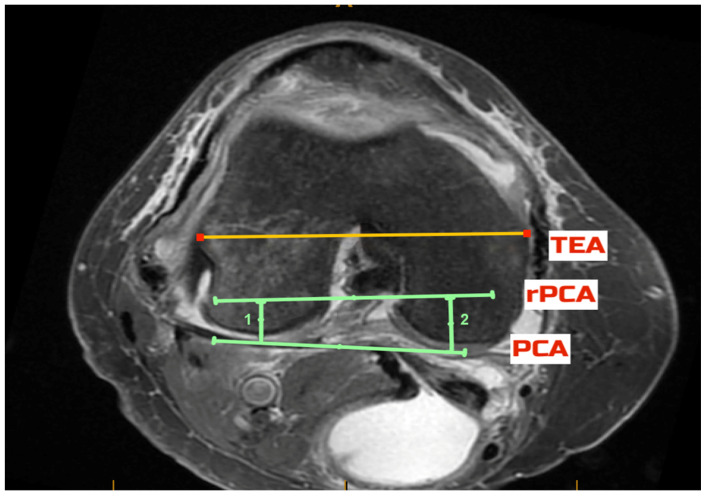
The transepicondylar axis (TEA—yellow line) was defined as the line connecting the most prominent medial and lateral aspects of the respective epicondyles (yellow line). The angles between the posterior condylar axis (PCA—lower green line) and the TEA (PCA/TEA angle), between the prosthetic posterior condylar axis (rPCA, top green line) and the PCA (rPCA/PCA angle) and between the rPCA and the TEA (rPCA/TEA angle) were measured in each case. The green lines 1 and 2 designate the posterior condylar resection depth at time of surgery.

**Table 1 jcm-11-06680-t001:** Patient-reported outcome measures: Western Ontario and McMaster Universities Osteoarthritis Index (WOMAC) and Knee Society Score (KSS). Values are presented as mean ± standard deviation.

	KSS	WOMAC
Pre-op	53.5 ± 7.5	45.8 ± 9.8
6 weeks	60.6 ± 10.0	22.9 ± 9.0
6 months	89.3 ± 10.6	3.4 ± 5.4
12 months	93.8 ± 4.5	0.9 ± 1.6
24 months	93.8 ± 6.2	0.4 ± 1.3

**Table 2 jcm-11-06680-t002:** TKAs having a reverse resection (higher resection at the affected compartment of the posterior femoral condyle). The resection (mm) as well as the posterior tibial slope of the medial, lateral tibia plateau and resected tibial surface are presented for each case. The thickness of the posterior condyles of the femoral component used in this study (CR, Attune TKA System, DePuy Synthes, Warsaw, IN) was 8 mm.

Predominantly Affected Compartment	Resection of Medial Posterior Condyle (mm)	Resection of Lateral Posterior Condyle (mm)	Medial Posterior Tibial Slope	Lateral Posterior Tibial Slope	Resected Posterior Tibial Slope	Cartilage Loss Medial	Cartilage Loss Lateral
lateral	7	8.5	0	7.2	7	1/3 loss	to bone
lateral	8	10	5	10	9	1/3 loss	to bone
lateral	7	8	5	6	6	no loss	1/3 loss
lateral	8	9	7	9	9	no loss	1/2 loss
medial	9	7	7	4	7	1/3 loss	no loss
medial	10	7	8.5	4	8	to bone	no loss
medial	12	10	10	7	9	1/2 loss	no loss
medial	11	9	9	9	9	2/3 loss	no loss

## Data Availability

The data that support the findings of this study are available from the corresponding author, upon reasonable request.

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
