# Peer review of "Femoral Component Rotation in Total Knee Arthroplasty Using a Tibia-First, Gap-Balancing, “Functional Alignment” Technique"

_jcm, 2022, doi:10.3390/jcm11226680_

Round 1
Reviewer 1 Report
Interesting study with appealing concept of addressing the difference in PTS during TKA. Because the tibial resection target matches the kinematic alignment target the authors term it kinematic alignment. However, I would suggest to use a different terminology, like: the tibia resection was performed to restore the patient pre-arthritic joint line. Please see comments in the PDF for further information.

Reviewer 2 Report
This study aims to describe femoral compartment rotation in TKA using a tibia first, gap balancing technique.
Overall the paper was an interesting and enjoyable read. There was a succinct introduction and analysis, with honest discussion and acknowledgement of study limitations.
It was reassuring to see there was no patellar maltracking determined and I would be genuinely curious to see the outcomes reviewed again at timepoints beyond two years.
Sentences 315 and 316 appear to have an extra space between some words (although may may be secondary to sentence justification). This is the only correction I would offer to this nice paper.
